# Achieving near-perfect light absorption in atomically thin transition metal dichalcogenides through band nesting

Seungjun Lee [1,5], Dongjea Seo[1,5], Sang Hyun Park[1], Nezhueytl Izquierdo[1], Eng Hock Lee [1], Rehan Younas [2], Guanyu Zhou[2], Milan Palei[2], Anthony J. Hoffman[2], Min Seok Jang [3], Christopher L. Hinkle[2], Steven J. Koester [1] ✉ & Tony Low [1,4] ✉

Near-perfect light absorbers (NPLAs), with absorbance, $\mathcal{A}$, of at least 99%, have a wide range of applications ranging from energy and sensing devices to stealth technologies and secure communications. Previous work on NPLAs has mainly relied upon plasmonic structures or patterned metasurfaces, which require complex nanolithography, limiting their practical applications, particularly for large-area platforms. Here, we use the exceptional band nesting effect in TMDs, combined with a Salisbury screen geometry, to demonstrate NPLAs using only two or three uniform atomic layers of transition metal dichalcogenides (TMDs). The key innovation in our design, verified using theoretical calculations, is to stack monolayer TMDs in such a way as to minimize their interlayer coupling, thus preserving their strong band nesting properties. We experimentally demonstrate two feasible routes to controlling the interlayer coupling: twisted TMD bi-layers and TMD/buffer layer/TMD tri-layer heterostructures. Using these approaches, we demonstrate room-temperature values of $\mathcal{A}$=95% at $\lambda$=2.8 eV with theoretically predicted values as high as 99%. Moreover, the chemical variety of TMDs allows us to design NPLAs covering the entire visible range, paving the way for efficient atomically-thin optoelectronics.

Near-perfect light absorbers (NPLAs), with absorbance, $\mathcal{A}$, of at least 99%, have a wide range of applications ranging from energy and sensing devices to stealth technologies and secure communications[1–7]. Among various types of NPLAs, those based on two-dimensional (2D) materials are of particular interest due to their potential for low-cost production of large-area atomically thin active layers. However, based on the simplest form of an optical resonator, the Salisbury screen[8], which is composed of a dielectric spacer and a metal reflector, the value of $\mathcal{A}$ for a monolayer 2D material is much lower than unity.

In the literature, improvement in $\mathcal{A}$ for 2D-based NPLAs devices is being pursued in two main directions: (1) utilizing complex plasmonic structures and (2) increasing the thickness of the 2D layers. In the former approach, by using plasmonic nanoparticles[9,10] and/or patterned metasurface structures, the $\mathcal{A}$ of atomically thin layers can be optimized up to near unity for a wide range of frequencies[11–19]. However, these complex structures require demanding nano-patterning processes resulting in expensive fabrication costs, and are thus limited to applications requiring only small-area structures. On the other hand, the use of thick 2D layers uses a much simpler device structure. For instance, 20 layers of transition metal dichalcogenides (TMDs), combined with conventional metal reflectors, such as Au or Ag, can exhibit near-perfect light absorption without any additional layer and

[1]Department of Electrical and Computer Engineering, University of Minnesota, Minneapolis, MN 55455, USA. [2]Department of Electrical Engineering, University of Notre Dame, Notre Dame, IN 46556, USA. [3]School of Electrical Engineering, Korea Advanced Institute of Science and Technology, Daejeon 34141, Republic of Korea. [4]School of Physics and Astronomy, University of Minnesota, Minneapolis, MN 55455, USA. [5]These authors contributed equally: Seungjun Lee, Dongjea Seo. ✉e-mail: skoester@umn.edu; tlow@umn.edu

patterned structure[20,21]. However, there is currently no feasible way of controllably growing such multilayer TMDs, and the ability to tune their peak absorption wavelength is severely limited.

In this work, we propose an approach based upon band nesting that allows NPLAs to be achieved in only two or three layers. It is known that optical absorption in monolayer TMDs benefits from an unusually strong band nesting effect[22,23], leading to optical conductivity of ~1 mS. This is roughly 10× higher than the value of graphene across the visible spectrum. If one supposes that the optical constant scales linearly with the number of layers, then theory would suggest that the ultimate TMD thickness for NPLAs with the Salisbury screen is just two atomic layers. However, the band nesting effect is disrupted by electronic coupling between layers. Here, we demonstrate that both twisted two-layer (2L) $MoS_2$ and three-layer (3L) $MoS_2$/graphene/$MoS_2$ heterostructures can effectively alleviate the interlayer coupling, allowing for the observation of substantial absorption enhancement up to 95%, but with the theoretical potential of $\mathcal{A}$ up to 99%. We further show that the same strategy, in principle, can be combined with other 2D materials to realize NPLAs covering the entire visible light range, which paves the way for practical applications of future atomically thin optoelectronics. Finally, we also demonstrate a viable pathway where the proposed TMD/buffer layer/TMD heterostructure can be grown in situ over a large area with no mechanical exfoliation and transfer.

## Results

### Ideal absorption with a single-mirror structure

We begin with a systematic theoretical exploration of NPLA design based on 2D materials. Between two dielectric media, the value of $\mathcal{A}$ for a material having a complex 2D optical conductivity of $\sigma_{2D}(\omega) = \sigma'(\omega) + i\sigma''(\omega)$ can be derived by the transfer matrix method and is described as[24]

$$\mathcal{A}(\omega) = \frac{4n_1\sigma'(\omega)Z_{VAC}}{|n_1 + n_2 + \sigma_{2D}(\omega)Z_{VAC}|^2}, \qquad (1)$$

where $n_1$ and $n_2$ are the refractive indexes on top and bottom sides of the 2D material, and $Z_{VAC} = 376.73\,\Omega = 1/\epsilon_0 c$ is the impedance of

vacuum. Since the maximum of $\sigma'$ coincides with a zero of $\sigma''$ due to the Kramers–Kronig relations, in the freestanding case ($n_1 = n_2 = 1$) the maximum $\mathcal{A}$ of a 2D material is simplified as a function of $\sigma'$, as shown in Fig. 1a. Without any optical cavity, the maximum $\mathcal{A}$ of the freestanding 2D layer is limited up to 0.5 when $\sigma' = 2\epsilon_0 c$.

To improve $\mathcal{A}$ beyond the freestanding limit, it is essential to utilize resonators[25,26]. Instead of using a complex structure, here, we consider the simplest optical resonator, which is the Salisbury screen consisting of a dielectric spacer and metal reflector[27]. Similar to the freestanding case, the value of $\mathcal{A}$ of the Salisbury screen can be calculated by the transfer matrix method (additional details are provided in Supplementary Sections S1, S2, and S3, and Supplementary Figs. S1 and S2). When the dielectric spacer thickness is optimized, the maximum $\mathcal{A}$ of the Salisbury screen can simply be described as

$$\mathcal{A}(\omega) = 1 - \frac{(1 - \sigma'(\omega)Z_{VAC})^2}{(1 + \sigma'(\omega)Z_{VAC})^2}. \qquad (2)$$

We have plotted $\mathcal{A}(\omega)$ in Fig. 1b. With this simple structure, $\mathcal{A}$ can approach 100% when $\sigma' = \epsilon_0 c$. One important lesson is that there is an optimal range of $\sigma'$ to realize NPLA. Comparing Fig. 1a, b, if $\sigma'$ ($\mathcal{A}$) of a freestanding 2D material ranges from $\sigma' = 2.17$ mS (41.2%) to $\sigma' = 3.24$ mS (47.1%,) we can expect $\mathcal{A} > 99\%$ using a single-mirror structure. Thus, 2D materials with these target conductivities would allow the realization of a nearly ideal Salisbury screen.

### 2D materials with high optical conductivity

Graphene has a universal optical conductivity of $\sigma' = e^2/4\hbar = 0.061$ mS ($\mathcal{A} \sim 2.3\%$) in the visible range due to its unique linear Dirac dispersion[28,29]. Therefore, unfortunately, graphene is not a suitable material for NPLA because its optical conductivity is an order of magnitude smaller than that required for the ultimate Salisbury screen. Among the family of 2D materials, TMDs exhibits strong light absorption due to its excellent nesting of electronic bands, which allows optical absorption over a broad region of the Brillouin zone at a specific wavelength[22,23]. Therefore, as a first step, we surveyed the $\sigma'$ value of the family of TMD monolayers which have finite bandgap and

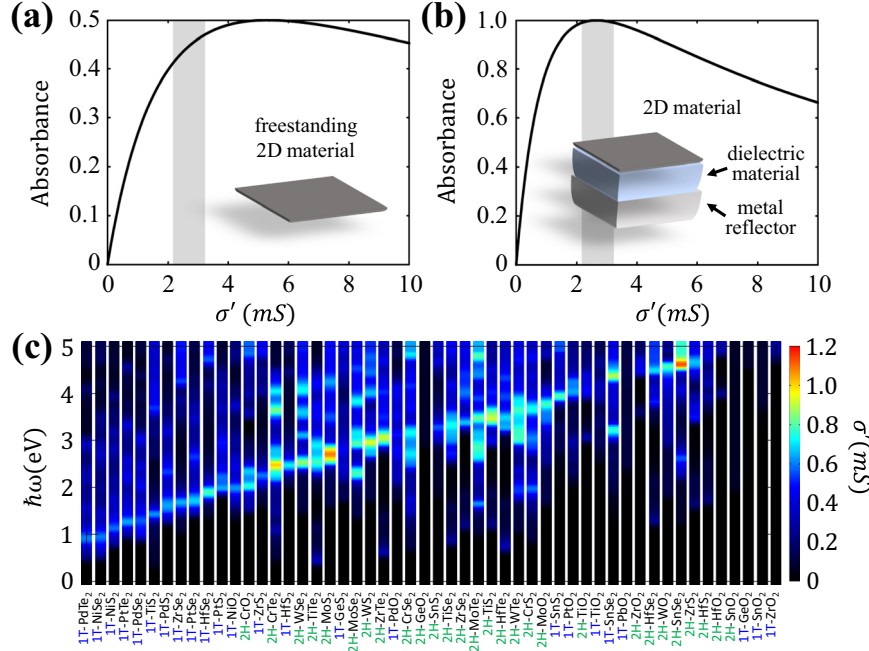

**Fig. 1 | Requirements of ideal absorption with a single-mirror structure.**
Maximum absorbance vs. the real part of the 2D optical conductivity $\sigma'$ of **a** a freestanding 2D material and **b** a 2D material with Salisbury screen. In both panels, the gray shaded areas highlight ideal absorption conditions (absorbance greater than 99% with Salisbury screen) where $\sigma'$ ranges from 2.17 mS to 3.24 mS. **c** Calculated $\sigma'$ of various freestanding 1L TMD materials obtained by first-principles calculations.

negative formation energy[30], by first-principles calculations based on the density functional theory (DFT), as shown in Fig. 1c. Among these TMD materials, the 2H phase of $MoS_2$ and $SnSe_2$ exhibit very high $\sigma'$ of around 1 mS, which is significantly larger than that of graphene. In particular, monolayer $MoS_2$ is readily accessible in experiments, in conjunction with the fact that cavity structures in the visible can be easily realized. However, unfortunately, they are still lower than the theoretical target for the ultimate Salisbury screen (2.17 mS < $\sigma'$ < 3.24 mS), implying that the realization of NPLA with TMD would require more than one layer.

### Degradation of band nesting due to the interlayer coupling

To enhance $\sigma'$ of a 2D material, the most straightforward method is to add an additional layer. We first proceed to calculate $\sigma'$ in 1L and 2L $MoS_2$. Figure 2a shows the calculated electronic structures of both 1L and 2L $MoS_2$ including spin−orbit couping (SOC). Due to the excellent band nesting ($\nabla_k(E_C - E_V) \sim 0$) along Γ-Q and Γ-M high symmetry lines (green arrows), 1L $MoS_2$ exhibits strong light absorption near 2.81 eV, where the absorption peak at this energy is often referred to as the C exciton[22,31]. Studies have shown that the excellent band nesting can be traced to the second nearest neighbor hopping between multiple metal d orbitals, which produces a local minimum in the conduction band at Q valley and thus promotes the nesting of the bands[32,33]. In 2L $MoS_2$, however, interlayer coupling splits the energies at the Γ valley of the valence band and the Q valley of the conduction band (red arrows), distorting and degrading the degree of band nesting. To visualize the relation between interlayer coupling and band nesting more clearly, in Fig. 2b, we represent the momentum-resolved energy difference between the highest valence bands and lowest conduction bands. In addition to the states along the high symmetry line, band nesting of 1L $MoS_2$ occurs over a large area of the first Brillouin zone, as indicated in yellow color. In 2L $MoS_2$, however, the yellow region noticeably shrinks, as interlayer coupling induces bonding and anti-bonding states between two layers, resulting in the degradation of the band nesting.

Based on the electronic structures, we calculated $\sigma'$ and $\mathcal{A}$ of both 1L and 2L $MoS_2$, and the results are shown in Fig. 2c, d, respectively.

Due to the strong band nesting, 1L $MoS_2$ exhibits a clear peak near 2.81 eV, and the calculated values of $\sigma'$ and $\mathcal{A}$ approach 0.96 mS and 25.5%, respectively. If the optical constant scales linearly with the number of layers, $\sigma'$ of 2L $MoS_2$ should be twice that of 1L $MoS_2$ (gray dashed line in Fig. 2c, d), which is very close to the requirement for NPLA.($\sigma'$ > 2.17 mS). However, in 2L $MoS_2$, the resonance peak due to band nesting splits into two smaller peaks, leading to maximum values of $\sigma'$=1.47 mS and $\mathcal{A}$ = 34 %, much lower than the target value.

It is worth noting that Fig. 2d does not contain the well-known A and B excitons of $MoS_2$[23] since these calculations consider only single-particle transitions. Excitonic effects can be incorporated through the Bethe-Salpeter equation, and these calculations were performed for 2L $MoS_2$ structures of varying interlayer distances (Supplementary Fig. S3). Here, we also observed the degradation of $\mathcal{A}$ with smaller interlayer distance. However, except for the appearance of the new A and B exciton features, the C exciton peak is only slightly red-shifted compared to its single-particle peak due to interband transitions. The C exciton is also accompanied by sharper linewidth, though it is sensitive to the excitonic lifetimes in these calculations. Since the C exciton serves to enhance this major peak responsible for the NPLA, in conjunction with minor redshift, we consider the more computationally efficient single-particle calculations to suffice for our purpose. For a similar reason, we also do not include SOC in what follows because it does not have critical effects on our main result (Supplementary Fig. S4).

### Absorption enhancement in decoupled twisted bilayer TMD

In order to understand how to improve $\sigma'$ in $MoS_2$ bilayers, we consider two practical approaches to reduce interlayer coupling: (1) layer twisting and (2) inserting a buffer layer between the top and bottom layers. These techniques are depicted in Fig. 3a. In van der Waals (vdW) materials, the ground-state stacking configuration has the minimum interlayer distance. Therefore, a finite twist angle leads to a larger interlayer distance and a weaker interlayer coupling. Especially in TMDs, the interlayer distance is minimized at 0° and 60° twist angles but is maximized near 30° due to the repulsion of the outermost chalcogen atoms[34,35]. We calculated $\mathcal{A}$ of freestanding twisted 2L $MoS_2$ with 21.81° and 32.22° twist angles, as represented in Fig. 3b.

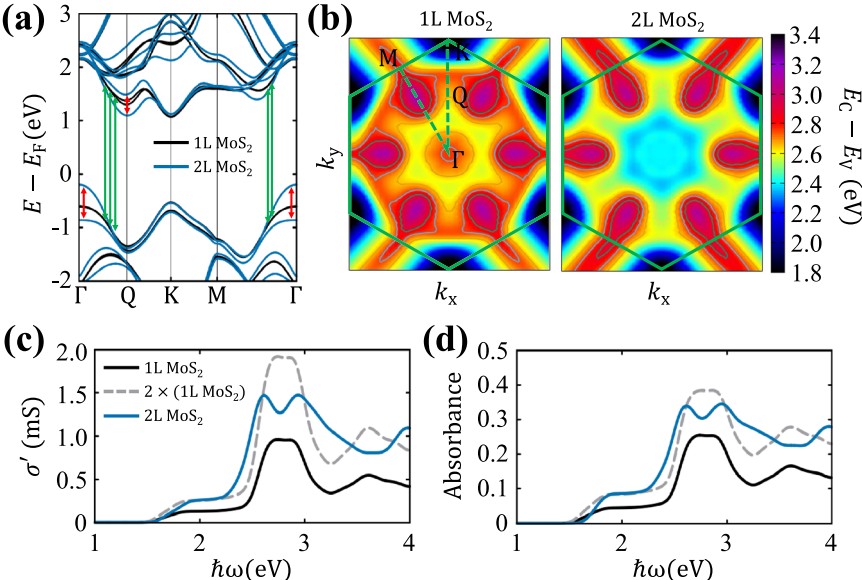

**Fig. 2 | Degradation of band nesting due to the interlayer coupling. a** Electronic band structures, **b** momentum-resolved band nesting maps, **c** real part of 2D optical conductivities, and **d** absorbances of freestanding mono and bilayer $MoS_2$. **a** Green arrows highlight an excellent band nesting of monolayer $MoS_2$, whereas red arrows show the destruction of the band nesting in the bilayer due to the interlayer coupling. **b** The color contour map indicates the energy difference between the energy of the lowest conduction, $E_C$, and the highest valence band, $E_V$, and the solid green lines show the first Brillouin zone. **c, d** A gray dashed line indicates an artificial bilayer structure having no interlayer coupling.

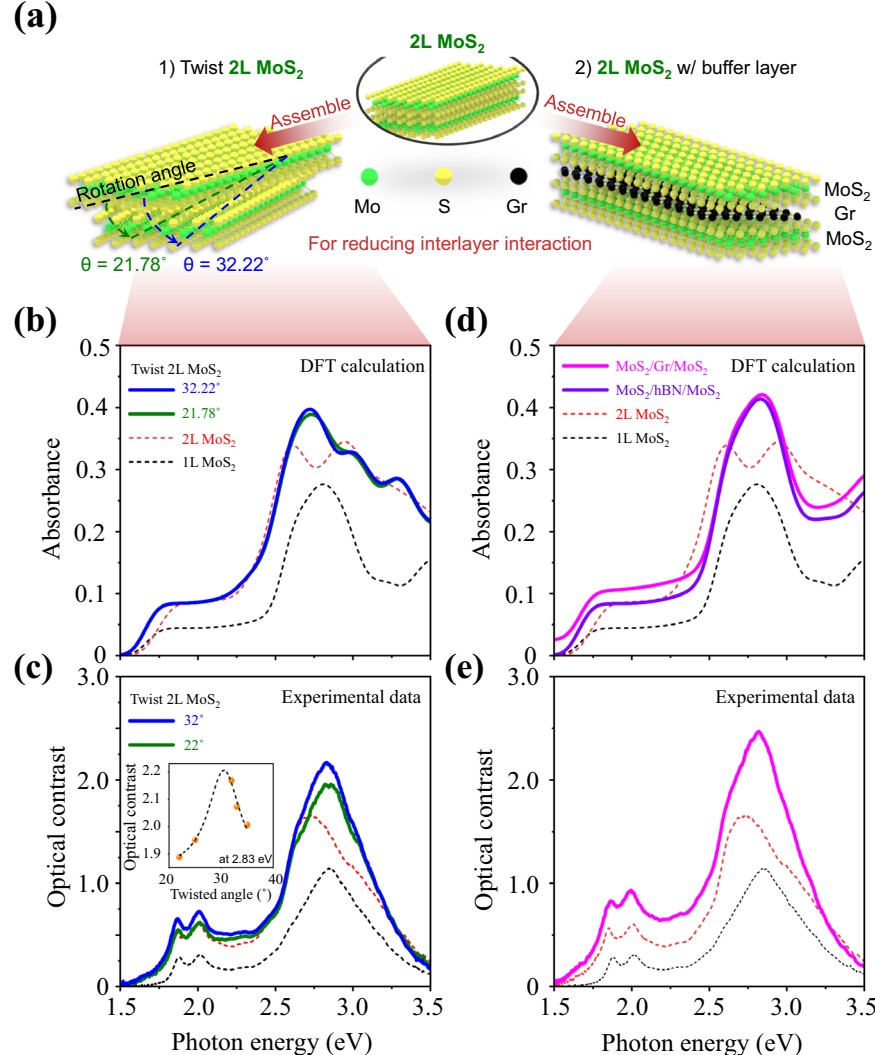

**Fig. 3 | Absorption enhancement through optimizing band nesting. a** Schematic diagram of absorption enhancement strategies; layer twisting and inserting a buffer layer between the top and bottom layers. Green, yellow, and black spheres indicate molybdenum, sulfur, and carbon atoms. Theoretically calculated absorbances and experimentally measured optical contrast of (**b**, **c**) twisted 2L MoS$_2$ and (**d**, **e**) 2L MoS$_2$ with an intermediate buffer layer. **b**–**e** We also show spectra of 1L and 2L MoS$_2$ shown as dashed black and red lines, respectively. **c** The inset shows the angle-dependent optical contrast of twisted 2L MoS$_2$ near the 30° twisted angle.

Interestingly, the maximum $\mathcal{A}$ of a freestanding twisted 2L MoS$_2$ approaches 39% and 40% with 21.81° and 32.22° twist angles, respectively, which are much higher than that of the value of 34% in the untwisted case (Fig. 2d). This result clearly shows the positive correlation between band nesting and interlayer distance. Moreover, in twisted bilayers, we observed that the single absorption peak is restored, suggesting reduced interlayer coupling.

To confirm our theoretical predictions, we performed optical measurements using a micro-reflectance setup, illuminating a selective sample area with a halogen white light source. Here, large-scale and high-quality 1L MoS$_2$ obtained by an Au-assisted exfoliation[36,37]. Further details are provided in Supplementary Sections S4 and S5 and Supplementary Figs. S5 and S6. Before exploring twist effects, first, we measured optical contrasts of 1L and 2L of MoS$_2$, which is directly proportional to the $\mathcal{A}$ of the film when using a transparent substrate[38,39] (Supplementary Section S6 and Supplementary Fig. S7). The optical contrast is defined as $(R-R_0)/R_0$ where R is the reflectance of the 2D material on the substrate and $R_0$ is the reflectance of the bare substrate. We clearly observed the well-known three excitonic peaks of MoS$_2$ corresponding to the A, B, and C excitons[23], which confirms the operation of our setup and the high quality of our samples. Also, a clear

redshift of the C exciton peak in the 2L MoS$_2$ shows evidence of the interlayer coupling.

The twisted 2L MoS$_2$ structures were fabricated using a standard dry-transfer polypropylene carbonate (PCC) stamp. To precisely control the twist angle, the MoS$_2$ flakes were picked up from the pre-patterned single crystal MoS$_2$ on a glass substrate and stacked to prepare the twisted bilayer. This procedure is described in more detailed in Supplementary Fig. S8 . The optical contrasts of 2L MoS$_2$ with twist angles of 22 and 32° are shown in Fig. 3c, which agree well with our theoretical calculations. When the twist angle goes from 0 to 32°, the peak position of the C exciton moves from 2.71 to 2.82 eV, corresponding to a blueshift of ~0.1 eV, indicating reduced interlayer coupling. Moreover, the optical contrast is maximized near 30° twist angle and tends to decrease as it moves away from 30°, which also clearly supports our hypothesis for the relation between the interlayer coupling and the twist angle. Additional experimental data are shown in Section S7 and Supplementary Fig. S9.

**Absorption enhancement in TMD/buffer layer/TMD structure**
Interlayer coupling can also be minimized by inserting a buffer layer between the TMD layers, as shown in Fig. 3a. Graphene (Gr) and

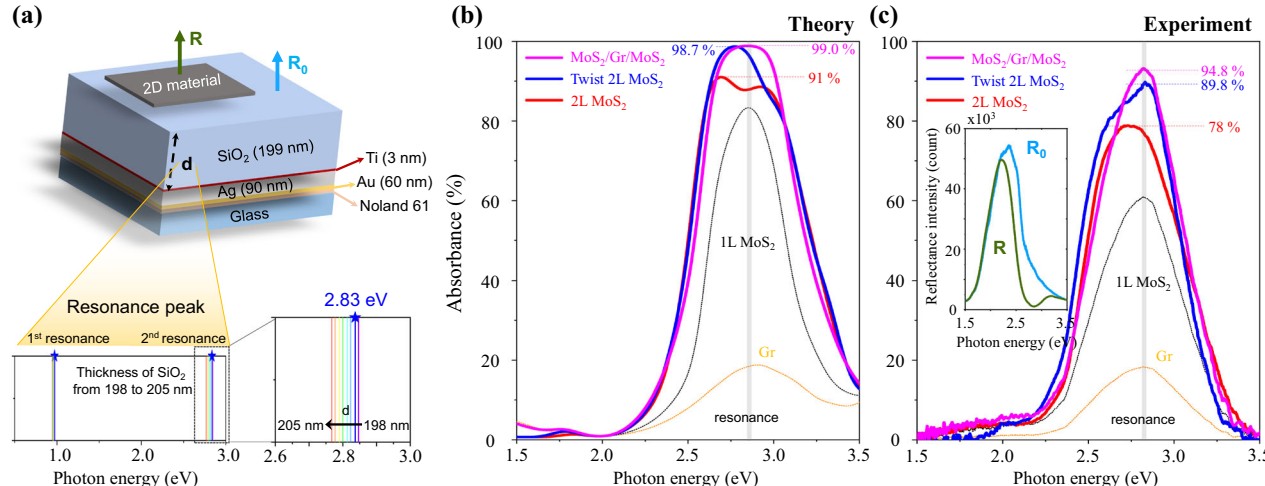

**Fig. 4 | Realization of near-perfect light absorbers with Salisbury screen. a** Schematic representation of our Salisbury screen structure. The thickness of the SiO₂ was 199 nm, satisfying the critical coupling condition for the photon energy of the C exciton (2.83 eV). **b** Calculated and **c** experimentally measured absorbance spectra of various MoS₂ structures with Salisbury screen. **c** Absorbance can be obtained from 1-R/R₀, where R and R₀ are the reflectances of the substrate with and without the 2D heterostructure, and the inset shows the raw data of reflectances for MoS₂/Gr/MoS₂ heterostructures.

hexagonal boron nitride (hBN) are used as buffer layer, primarily due to their experimental feasibility. Optical absorption measurements and first-principles calculations both confirmed that they indeed served as good buffer layers. Figure 3d shows the calculated $\mathcal{A}$ value of MoS₂/hBN/MoS₂ and MoS₂/Gr/MoS₂ heterostructures. In contrast to 2L MoS₂, we observed a single C exciton peak in both MoS₂/hBN/MoS₂ and MoS₂/Gr/MoS₂ heterostructures, implying that the band nesting of the MoS₂ layers is preserved. Interestingly, both $\mathcal{A}$ of MoS₂/hBN/MoS₂ and MoS₂/Gr/MoS₂ have calculated $\mathcal{A}$ values (41.4% and 42.0%, respectively), which are not only higher than twisted 2L MoS₂ but also the near-perfect absorption condition of $\mathcal{A} = 41.2\%$. Also, we confirmed that both MoS₂/hBN/MoS₂ and MoS₂/Gr/MoS₂ have almost exactly 2 × optical conductivity of 1L MoS₂, implying effective suppression of interlayer coupling. Additional details of these calculations are provided in Supplementary Fig. S10.

Similar to the twisted case, the MoS₂/Gr/MoS₂ heterostructure was fabricated by a three-step transfer process, as shown in Supplementary Fig. S11. The optical reflectance measurements were then performed and the results are shown in Fig. 3e. For this heterostructure, $\mathcal{A}$ slightly increases due to the small but finite absorption of Gr while the excitonic peak positions remain unchanged compared to those of MoS₂ due to weak interlayer interaction. This is further confirmed by the $E_{2g}^1$ and $A_{1g}$ modes of Raman spectra. Additional details are provided in Supplementary Section S8 and Supplementary Fig. S12. More interestingly, the optical contrast of the C exciton in the MoS₂/Gr/MoS₂ heterostructure was remarkably enhanced by 116% and 48% compared to those of 1L and 2L MoS₂, and the redshift is not observed. All these observations are in excellent agreement with our theoretical predictions.

For more complete validation beyond a specific material set, we also realized WSe₂/ZnSe/WSe₂ heterostructures grown by molecular beam epitaxy, where Supplementary Section S9 provides more details of the material growth. This demonstration is important in that it shows the capability of all in situ and wafer-scale growth of NPLA heterostructures, without the need for exfoliation and layer transfer. In addition, the ZnSe is a cubic 3D material in the zincblende phase, illustrating that the decoupling interlayer need not be a 2D material or even hexagonal. Interestingly, as shown in Supplementary Fig. S13, we also observed similar $\mathcal{A}$ enhancement. Therefore, the theoretical and experimental results undoubtedly demonstrate that an atomic buffer layer can be an effective strategy to $\mathcal{A}$ enhancement.

## Realization of NPLAs with Salisbury screen

Figure 4a illustrates the geometry of the proposed twisted 2L MoS₂ and MoS₂/buffer layer/MoS₂ heterostructures with Salisbury screen structure composed of SiO₂ and Ag reflector. Here, the thickness of the SiO₂ was chosen to be 199 nm, satisfying the critical coupling condition for the photon energy of 2.83 eV as described in Supplementary Section S2. We theoretically calculated $\mathcal{A}$ of various MoS₂ structures, as represented in Fig. 4b. Due to resonance effects, near the C exciton peak, $\mathcal{A}$ clearly enhances even in 1L Gr and 1L MoS₂. More interestingly, the theoretical maximum $\mathcal{A}$ of 2L MoS₂ is calculated to be 91.0%, which can be further enhanced to 98.7% by using 32.22° twist angle. Finally, we calculate that $\mathcal{A}$ can increase up to 98.8% and 99.0% by using hBN and graphene buffer layers, respectively.

To validate our theoretical prediction by experiments, we fabricated SiO₂(199 nm)/Ti(3 nm)/Ag(90 nm)/Au(60 nm) heterostructure on a glass substrate by the template stripping method (Supplementary Fig. S14), which offers a ultra-flat surface morphology (Supplementary Fig. S15). Then, we performed optical measurements, as shown in shown in Fig. 4c. In this structure, almost all incident light (for visible) is reflected by the Ag mirror at 2.83 eV, as shown in the measured reflectance R, as displayed in the inset of Fig. 4c. $\mathcal{A}$ can be obtained from 1-R/R₀, where R and R₀ are the reflectances of the substrate with and without the 2D heterostructure, respectively, as described in Fig. 4a. Consistent with our calculation, we also observed a strong resonance peak even with just Gr, validating the precisely controlled thickness of the SiO₂ layer. The maximum absorbance of pristine 2L MoS₂ is measured to be ~78% with a mild redshift, which is a slightly lower value than that of theory, but generally in good agreement. With a nonzero twist angle, $\mathcal{A}$ has reached almost 89.8% at the resonance peak of 2.83 eV with 28° twist angle, and it weakens when the rotation angle is further away from 30° ($\mathcal{A}$=86.3% at 26°, 84.5% at 23°). Supplementary Fig. S16 provides additional details of these measurements. Moreover, $\mathcal{A}$ in the MoS₂/Gr/MoS₂ is even higher than that of twisted 2L MoS₂, and reaches up to 94.8%, a remarkably high value for structure with thickness <2 nm. These results are a strong confirmation that reducing interlayer coupling is a very efficient and feasible strategy to enhance $\mathcal{A}$ in the multilayer TMDs, showing that a simple and cost-effective NPLAs (or perfect light absorbers) can be realized by combining 2D materials and a single-mirror cavity structure. Additional discussions on the multilayer TMDs are provided in Supplementary Section S10, Supplementary Figs. S17 and S18.

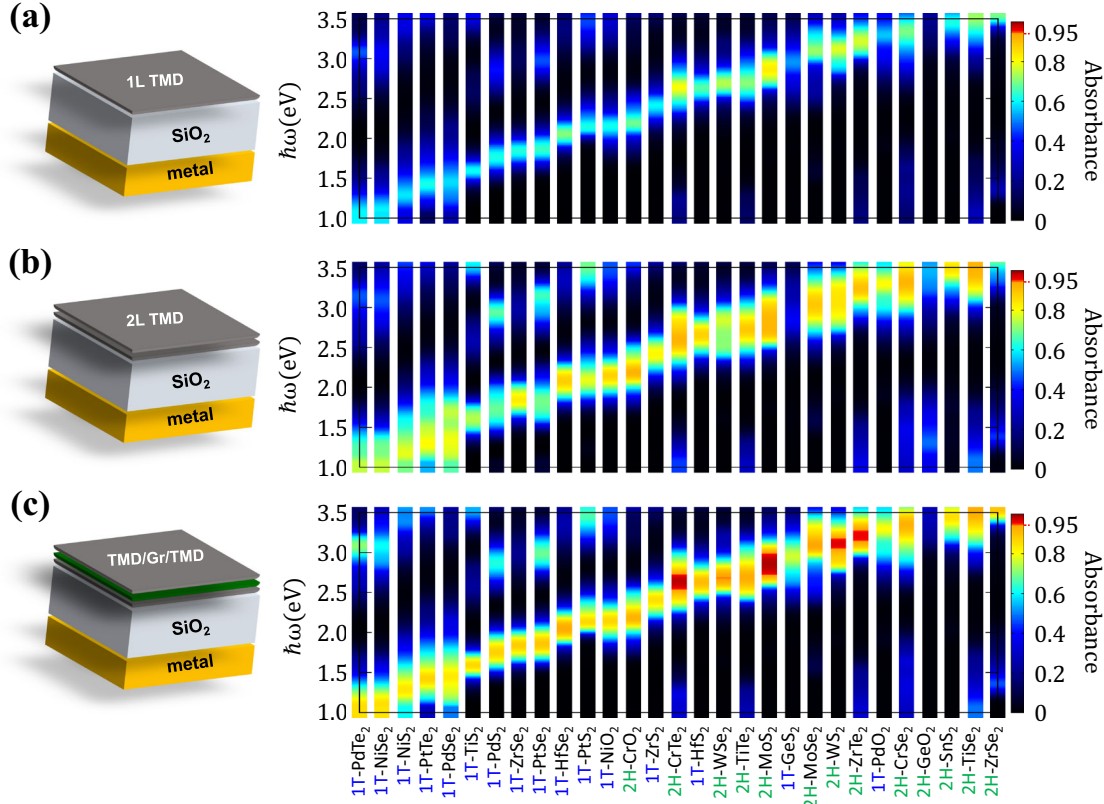

**Fig. 5 | Near-perfect light absorbers over a wide frequency.** Calculated absorbance spectra on a SiO2/Ag cavity structure of various 2D materials with **a** 1L, **b** 2L, and **c** 2L with buffer. We note that the thickness of the SiO₂ was optimized for each material, as summarized in Supplementary Fig. S19.

## NPLAs over a wide frequency by using various TMD materials

One of the drawbacks of many optical resonance systems is that they require precisely designed complex structures depending on the optimum wavelength. However, in our approach, optimum wavelength can be tuned by simply optimizing the thickness of the dielectric layer, and thus can be easily adapted to the other material systems. To expand our idea to a wider family of materials, we investigated the optical conductivities and $\mathcal{A}$ of various 2D materials, whose operating wavelength appears in the visible region. Figure 5a shows the calculated $\mathcal{A}$ of a total of 29 1L 2D materials embedded in the Salisbury screen, which cover the entire visible light spectrum. In the 1L limit, the highest $\mathcal{A}$ value is calculated to be 82.5% for MoS₂, followed by 2H-CrTe₂ (79.9%), and 2H-WS₂ (77.7%). Using a 2L structure, as shown in Fig. 5b, we obtained higher $\mathcal{A}$ for all materials, but these values are still appreciably lower than unity. Finally, as shown in Fig. 5c, when an intermediate buffer layer is used, we clearly observed enhanced $\mathcal{A}$ for most materials, implying that reducing interlayer coupling generally increases $\mathcal{A}$ of a wide range of 2D materials. In addition to MoS₂, the other materials exhibit the strong absorption are 2H-CrTe₂ (97.9%), 2H-WS₂ (96.4%), and 2H-ZrTe₂ (95.8%). Therefore, the chemical variety of 2D materials opens a possibility to realize atomically thin NPLAs for the entire visible light range.

## Discussion

In summary, through harnessing the full effect of band nesting, we demonstrated the ultimate Salisbury screen with only two uniform atomic layers of TMDs. Our first-principles calculations revealed that interlayer coupling can have a detrimental effect on the degree of band nesting. To optimize band nesting and enhance absorption, we prepared twisted 2L MoS₂ and MoS₂/Gr/MoS₂ heterostructure and observed strong absorption enhancement. Through this strategy, without any complex optical structures, we realized NPLA with

absorption as high as 95% using only simple single-mirror reflector structure. We further confirmed that our strategies for optimizing band nesting are widely adaptable to the other 2D materials, offering an attractive and scalable platform for NPLAs across the visible spectrum.

## Methods
### Computational details

We performed first-principles calculations based on density functional theory[40] as implemented in Vienna ab initio simulation package (VASP)[41]. The electronic wavefunctions were expanded by plane wave basis with kinetic energy cutoff of 400 eV. We employed the projector-augmented wave pseudopotentials[42,43] to describe the valence electrons, and treated exchange-correlation (XC) functional within the generalized gradient approximation of Perdew−Burke−Ernzerhof (PBE)[44]. We employed Grimme-D3 method[45] to describe van der Waals interaction. To mimic layered structure in periodic cells, we included a sufficiently large vacuum region in-between neighboring cells along the out-of-plane direction. The supercell structures for twisted 2L MoS₂ and MoS₂/Gr(or hBN)/MoS₂ heterostructures were constructed by the coincidence lattice method[46,47] to minimize artifical strain within reasonable cell size (Supplementary Fig. S20). The Brillouin zone were sampled using 21 × 21 × 1, 15 × 15 × 1, and 9 × 9 × 1 *k*-point meshes for unitcell, MoS₂/Gr(or hBN)/MoS₂ heterostructure and twisted 2L MoS₂, respectively.

The frequency-dependent optical conductivity $\sigma(\omega)$ was calculated by Kubo−Greenwood formula defined as[48,49]

$$\sigma_{\alpha\beta}(\hbar\omega) = \frac{ie^2\hbar}{N_k\Omega_c}\sum_{\mathbf{k}}\sum_{n,m}\frac{f_{m\mathbf{k}}-f_{n\mathbf{k}}}{\varepsilon_{m\mathbf{k}}-\varepsilon_{n\mathbf{k}}}\frac{\langle\psi_{n\mathbf{k}}|v_\alpha|\psi_{m\mathbf{k}}\rangle\langle\psi_{m\mathbf{k}}|v_\beta|\psi_{n\mathbf{k}}\rangle}{\varepsilon_{m\mathbf{k}}-\varepsilon_{n\mathbf{k}}-(\hbar\omega+i\eta)},$$

where $\alpha,\beta$ are Cartesian directions, $N_k$ is the number of k-points, $\Omega_c$ is the volume of the cell structure including vacuum region. Physically, the quantitative value of $\sigma'$ strongly depends on the carrier relaxation time contributed from various scattering sources and temperatures. To deal with this quantity in the Kubo formula, we used a constant broadening parameter of $\eta$ chosen to be 100 meV, which guarantees good agreement with previous reports (Supplementary Fig. S21)[24]. The numerical calculations for $\sigma(\omega)$ were performed by Wannier90 package[50–52]. For the wannierization, we used $d$ orbital projection for most transition metals, $s$ and $p$ orbital projections for chalcogenides, and $s$, $p$, and $d$ orbital projections for Zr, respectively. The 2D optical conductivity $\sigma_{2D}(\omega)$ was calculated as $\sigma_{2D}(\omega) = \sigma(\omega)L$, where $L$ is the out-of-plane lattice constant of periodic cell structure including the vacuum region.

### Fabrication of large-scale MoS₂ by Au-assisted exfoliation method

At the first step, a highly polished (111) bare Si wafer was deposited with a 150-nm thick Au film as an ultra-flat template by e-beam evaporation (CHA industries, SEC 600). On top of the Au layer, polymethyl methacrylate (MicroChem, 950 PMMA C4) as protecting layer was spin-coated at a rate of 1000 rpm for 60 s. After baking at 120 °C for 2 min, the Au layer is peeled off from the Si substrate with blue tape. Then a freshly cleaved layered bulk $MoS_2$ was gently pressed on a freshly Au film to establish large-scale $MoS_2$/Au contact. The several hundred microns lateral monolayer $MoS_2$ is peeled off due to the interaction between Au and the sulfur atom of $MoS_2$, which is stronger than the interlayer vdW interaction in bulk $MoS_2$. The template-stripped Au can exfoliate and transfer the $MoS_2$/Au film/PMMA/blue tape to the desired substrate and then dipped in the acetone to dissolve the PMMA to peel off the blue tape. Acetone leaves a residue thus rinse with 2-propanol to remove the residue. Finally, the top Au film was removed by an aqueous $KI/I_2$ etchant (Sigma Aldrich, "Au etchant, standard") to release the $MoS_2$. To wash off Au etchant residues, the $MoS_2$ on the Si substrate was rinsed with acetone and 2-propanol and blown dry with nitrogen. More details are provided in Supplementary Section S4.

### Fabrication of Salisbury screen

For the Salisbury screen structure with $SiO_2$ as an optical spacer, a bare silicon wafer was first heated on a hot plate at 180 °C for 5 min and then surface treatment by oxygen plasma (Advanced vacuum, Vision 320) for 5 min. The $O_2$ plasma cleaning process can remove organic residues and other contaminants that may be present on the surface of the Si wafer, resulting in a clean and uniform surface for subsequent processing steps. The silicon wafer was deposited with a 60-nm-thick Au film as a sacrificial layer by an electron-beam evaporator (CHA Industries, SEC 600). A $SiO_2$ as a dielectric spacer was deposited at 199 nm thick by plasma-enhanced chemical vapor deposition (Plasma-Therm, PECVD). An Ag and Au film were deposited at 90 nm and 60 nm thick, respectively, by electron-beam evaporation (CHA industries, SEC 600). Next, we used the template stripping method[53,54] using a photocurable epoxy (Norland, NOA 61) as an adhesive to transfer the entire cavity structure to a glass substrate. Finally, the sacrificial layer of the top Au film was removed using Au etchant (Sigma Aldrich) to obtain a flat surface of $SiO_2$ on the bottom Ag mirror. More details are provided in Supplementary Fig. S14.

### Characterization of the structure

Raman measurements were performed at room temperature using a confocal Raman spectroscopy (Witec Alpha 300R) under the excitation of a 532 nm laser. The optical characterization of the twisted bilayer Mo₂ and MoS₂/Gr/MoS₂ heterostructure have been carried out with optical contrast spectroscopy. The optical contrast spectrum was acquired from a self-built system equipped with a reflective light configuration. We collect the light from an area of the sample using a white light source (e.g., the halogen light source).

### Data availability

Data supporting key conclusions of this work are contained within the paper and Supplementary Information. All raw data used in the current study are available from the corresponding author under request.

### Code availability

All first-principles calculations were performed by VASP and Wannier90 codes. Additional scripts used in the current study are available from the corresponding author under request.

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

## Acknowledgements

This work was primarily supported by the National Science Foundation (NSF) through the DMREF program under Award No. DMR-1921629 and DMR-1921818, and in part by the NSF under Award No. ECCS-1542202. Portions of this work were conducted in the Minnesota Nano Center, which is supported by the NSF through the National Nanotechnology Coordinated Infrastructure (NNCI) under Award No. ECCS-2025124. S.L. is also supported by Basic Science Research Program through the National Research Foundation of Korea (NRF) funded by the Ministry of Education (NRF-2021R1A6A3A14038837). M.S.J. acknowledges the support by the NRF grant funded by the Korea government (MSIT) (2022R1A2C2092095). M.S.J. and T.L. thank Sangjun Han for his con-tribution to calculating the optical absorption in 2D material-based Salisbury screens. D.S. and S.J.K. thank Sang-Hyun Oh for use of the optical setup used in this work.

## Author contributions

S.L., D.S., S.J.K., and T.L. conceived this study. S.L. performed the first-principles calculations with the help of E.H.L. and T.L. D.S. fabricated and characterized the devices, performed the optical measurements with the help of N.I. and S.J.K. S.H.P. performed model calculations on the optical properties of 2D thin films with technical inputs from M.S.J. and T.L. R.Y., G.Z., and C.L.H. did MBE growth for 2D TMD materials and $WSe_2/ZnSe/WSe_2$ heterostructures. M.P. and A.J.H. did some of the optical measurements. All authors contributed to the discussion on the results. S.L., D.S., S.J.K., and T.L. wrote the manuscript with the input from all authors. S.J.K. and T.L. supervised the whole project.

## Competing interests

The authors declare no competing interests.
