## [Peer Review File · Nature Communications]

Reviewers' Comments:

Reviewer #1:

Remarks to the Author:

Lee et al has reported a systematic study on the optical absorption enhancement of 2D semiconductors by both band nesting and mirror reflection architecture design. The novelty of this work lies in the systematic optical conductivity of various 2D materials with the most intense peaks covering infrared to ultraviolet regions. However, there are still a few very important issues that are poorly understood:

1. Page 5 line 14-15 and Fig.2a, how do the authors define the band nesting? The green arrows? It is quite different from the definition in Phys. Rev. B 2013 and Nat. Commun. 2014 by Neto et al. Do the authors really understand?
2. Page 5 line 22-23 and Figure 2b, "band nesting of 1L MoS₂ occurs over a large area of the first Brillouin zone, as shown by the red color." Here the red color regions in Fig.2b are actually maximums in the energies of E_c-E_v , but they are not band nesting regions. The right one should be the hexagonal yellow rings (local energy minimum) in the 1L case, almost the midpoint of γ -Q. BSE calculation by Qiu et al PRL2015 also support the k-space excitonic wavefunction at the midpoint of γ -Q. In 2L case, the light blue hexagonal rings correspond to the band nesting.
3. Fig. 4a, the 1st resonance peak at 1.0eV has almost the same height (intensity) compared to 2nd resonance peak at 2.8eV, but this is contradictory with the experimental observation that no obvious absorption appear below the A excitation. This is not self-consistent.
4. For Fig.2b, use simple color bars (red-to-black), not rainbow or spectrum, to better visualize the energy minima.
5. Page 9 line 16, "Interlayer coupling can also be minimized by inserting a buffer layer", insert?
6. Page 10 line 20, "the thickness of the SiO₂ thickness" ?
7. Page 14 line 17 methods part, For the calculations, both d and p orbitals of Mo should be taken into account. Did the authors compare the cases with and without Mo-p orbitals?
8. Graphene is used as a buffer layer, but graphene is a half metal and induces remarkable quenching effect. why graphene is chosen?
9. BN and graphene play similar roles in reducing the interlayer coupling and in enhancing optical absorption. So any buffer layer made of any other material also works?
10. Are there experimental results from 3L and 4L, to compare with 2H-bilayer and twisted bilayer? And then 3L with interlayer twist angles?

Therefore I cannot support its publication at this stage, considering the limited novelty.

Reviewer #2:

Remarks to the Author:

The manuscript by Seungjun Lee et al. presents interesting results of theoretical and experimental work aiming toward achieving near-perfect light absorption in atomically thin transition metal dichalcogenides through band nesting and embedding these materials in an optical resonator. The nesting of valence and conduction bands in TMDCs i.e. existence of parallel in energy bands is visible in results of ab-initio band structure calculation by several groups including the author, resulting in large optical conductivity. The large optical conductivity from Kohn Sham orbitals is also partially supported by BSE-GW methodology. The band nesting is significant for a single layer. For highly absorbing materials one needs to increase the number of layers. Unfortunately as argued by the authors interlayer coupling is reducing band nesting effects. Hence the authors propose to increase the number of layers while reducing their electronic coupling by either twisting two layers or inserting a different material in between the TMD layers. In the end the layers of TMDCs are inserted into Salisbury screen resonator and very high absorptivity is demonstrated. Overall I find this work to be a very nice application of unique properties of TMDCs – band nesting. The paper is well written and results appear sound. Where the paper could improve is to provide not only observation but also explanation of the physical origin of band nesting. This has been done and as an example I refer authors to two papers: Bieniek et al., "Band nesting, massive Dirac Fermions and Valley Lande and Zeeman effects in transition metal dichalcogenides: a tight

binding model", Phys.Rev.B 97, 085153 (2018). and Bieniek, et al. , "Band nesting and exciton spectrum in MoS₂", Phys.Rev.B101, 125423 (2020).

This additional discussion would significantly improve the quality and impact of the manuscript and I would be happy to recommend publication.

Reviewer #3:

Remarks to the Author:

Authors used band nesting of TMDC materials and Salisbury geometry to achieve NPLA. They did thorough theoretical study on absorbance and optical conductivity of different types of 1L 2D materials. They also provided simulation, detail calculation, material and device fabrication, and characterization. Overall, this piece of work is interesting and could be useful for 2D materials community. However, authors need to address below queries and correct the mistakes.

1. In the introduction, they mentioned that "these complex structures require demanding nano-patterning processes resulting in expensive fabrication costs and are thus limited to applications requiring only small-area structures". In their twisted 2L MoS₂ fabrication, E-beam patterning is involved. This is a slow process and cannot be expand to large area. How will the authors justify the statement mentioned in introduction?

2. Based on the findings from Figure 1(c), 2H phase of SnSe₂ has the highest σ' of more than 1mS. Why did the authors choose MoS₂ instead of SnSe₂ to demonstrate?

3. The authors did not discuss on full width at half maximum (FWHM) of respective curves in Figure 4(c). The twist 2L MoS₂ has the broader spectrum among others. Why?

4. When the authors obtained absorbance from different samples with reflection measurement, did they consider scattering effect? Can you include the experiment result to check the scattering responses from your samples?

5. At page 12, it should be "conclusion" or "summary" instead of "discussion".

6. At page 15, during the fabrication of Salisbury screen structure, a bare silicon wafer without pre-cleaning was used. Do you think it is not necessary? Why?

7. For Salisbury screen structure, what is the tolerance of each layer thickness such as 199nm of SiO₂, 90nm of Ag and 60nm of Au?

8. During characterization of samples, did the author take into consideration of polarization effect especially on twisted sample?

9. The indication of PMMA and Au is not correct at step 4 of Fig S4 (a).

10. In page 8 of supporting information, there is typo error. "tow" should be "two".

11. In page 14 of supporting information, the abbreviation of graphene should be "Gr" instead of "gr".

12. In S9, the authors used 192nm of SiO₂, 3 nm of Titanium and 100 nm of Silver to form Salisbury screen structure. Do all layers need to have the exact thickness as mentioned? Why is there layers difference in this Salisbury screen structure from MoS₂ sample?

REVIEWER COMMENTS

Reviewer #1 (Remarks to the Author):

Lee et al has reported a systematic study on the optical absorption enhancement of 2D semiconductors by both band nesting and mirror reflection architecture design. The novelty of this work lies in the systematic optical conductivity of various 2D materials with the most intense peaks covering infrared to ultraviolet regions. However, there are still a few very important issues that are poorly understood:

We thank the Reviewer for his/her time and effort in reviewing our manuscript. Also, we appreciate important and constructive comments and suggestions, which have helped us improve the manuscript. Please find below our point-by-point replies.

1. Page 5 line 14-15 and Fig.2a, how do the authors define the band nesting? The green arrows? It is quite different from the definition in Phys. Rev. B 2013 and Nat. Commun. 2014 by Neto et al. Do the authors really understand?

Thank you for your careful review and raising this important issue. As Reviewer #1 pointed out, the band nesting area of 1L MoS₂ in the [Phys. Rev. B 88, 115205 (2013)] and [Nat. Commun. 4543 (2014)] are slightly different from ours because they included spin-orbit coupling (SOC), but we did not. Without SOC, as shown in the previous version of Fig.2(b), you can clearly find “red” region which satisfying band nesting condition ($\nabla_k(E_c - E_v) \sim 0$, defined by [Phys. Rev. B 88, 115205 (2013)]) and corresponding to the green arrows in the previous version of Figure 2(a).

With the inclusion of SOC, small spin splitting occurs near K and Q valleys, resulting in small deviation of band nesting. However, in our previous manuscript, we didn't include SOC effects because it does not have critical effects on our main result: the relation between band nesting vs interlayer interaction, as shown in Fig. R1.

Fig. R1. Effects of SOC on real part of 2D optical conductivities of 1L and 2L MoS₂.

Although SOC has minor effects on our main physics, its omission in the manuscript could confuse the reader. Therefore, in our revised manuscript, we replaced our Figure 2 with the results including SOC and also revised the text accordingly.

For the cases of the twisted bilayer and MoS₂/Graphene/MoS₂ heterostructures, SOC will not be included because (1) it does not have critical impact on our work and (2) they require expensive computational costs. For example, for the 32.22° twisted bilayer case, we have to use a large supercell structure (lattice constant = 11.48 Å) and relatively dense k-grid (9x9x1) to get converged results. To justify our choice, in our revised manuscript, we also added a few sentences and added Fig. R1 in the Supplementary Information as Figure S4.

Figure 2 on page 6,

Previous version of Figure 2

Revised Figure 2

Second paragraph on page 5,

Figure 2(a) shows the calculated electronic structures of both 1L and 2L MoS₂ including spin-orbit coupling (SOC). Due to the excellent band nesting ($\nabla_k(E_c - E_v) \sim 0$) near the Γ and Q valleys along Γ -Q and Γ -M high symmetry lines (green arrows), 1L MoS₂ exhibits strong light absorption near 2.81 eV, where the absorption peak at this energy is often referred to as the C exciton.^{22,31}

Second paragraph on page 5,

In addition to the states along the high symmetry line, band nesting of 1L MoS₂ occurs over a large area of the first Brillouin zone, as shown by the red-indicated in yellow color. In 2L MoS₂, however, the red-yellow region noticeably shrinks, as interlayer coupling induces bonding and anti-bonding states between two layers, resulting in the degradation of the band nesting.

Second paragraph on page 6,

Due to the strong band nesting, 1L MoS₂ exhibits a clear single-peak near 2.81 eV, and the calculated values of σ' and A approach 1.090.96 mS and 27.725.5 %, respectively. If the optical constant scales linearly with the number of layers, σ' of 2L MoS₂ should be twice that of 1L MoS₂ (gray dashed line in Fig. 2(c) and (d)), which satisfies is very close to the requirement for NPLA. ($\sigma' > 2.17$ mS). However, in 2L MoS₂, the resonance peak due to band nesting splits into two smaller peaks, leading to maximum values of $\sigma' = 1.551.47$ mS and $A = 34$ %, much lower than the target value.

Second paragraph on page 7,

Since the C exciton serves to enhance this major peak responsible for the NPLA, in conjunction with minor redshift, we consider the more computationally efficient single-particle calculations to suffice for our purpose. For the similar reason, we also do not include SOC in what follows because it does not have critical effects on our main result; the relation between band nesting and interlayer interaction. (Fig. S4 in SI.)

Figure S4 in Supplementary Information,

2. Page 5 line 22-23 and Fig. 2b, “band nesting of 1L MoS₂ occurs over a large area of the first Brillouin zone, as shown by the red color.” Here the red color regions in Fig.2b are actually maximums in the energies of $E_c - E_v$, but they are not band nesting regions. The right one should be the hexagonal yellow rings (local energy minimum) in the 1L case, almost the midpoint of gamma-Q. BSE calculation by Qiu et al PRL2015 also support the k-space excitonic wavefunction at the midpoint of gamma-Q. In 2L case, the light blue hexagonal rings correspond to the band nesting.

We are in total agreement with Reviewer #1. The correct region of band nesting of 1L MoS₂ (including SOC) should be midpoint of the Γ -Q line. As we explained in comment #1, this difference is due to SOC. As you can see below Fig. R2, if we include SOC, we also can find a large yellow hexagonal ring between Γ -Q line, which agree well with previous papers. As we already answered in the previous comments, we revised Figure 2 and included SOC effects in the revised manuscript.

Fig. R2. Effects of SOC on band nesting region of 1L MoS₂.

3. Fig. 4a, the 1st resonance peak at 1.0eV has almost the same height (intensity) compared to 2nd resonance peak at 2.8eV, but this is contradictory with the experimental observation that no obvious absorption appears below the A exciton. This is not self-consistent.

It is important to point out that the graph in Fig. 4a simply indicates the resonance positions of our optical cavity structure, which has 199-nm SiO₂ above a metallic mirror. The cavity has two resonances between 0 and 3 eV occurring at 1.0 eV and 2.83 eV. However, critical absorption always requires two conditions: (1) an optical resonance of the Salisbury screen, and (2) the proper value of optical conductivity (Fig. 1(b)). The 1st resonance region at 1 eV does not satisfy condition #2 since $\sigma' = 0$ at 1 eV photon energy is less than the band gap of MoS₂, and thus there is no absorption. For these reasons, we believe our results are indeed self-consistent.

4. For Fig.2b, use simple color bars (red-to-black), not rainbow or spectrum, to better visualize the energy minima.

We thank the reviewer for this suggestion, and made these new plots as suggested by referee in Fig. R3. However, we believe that our current Fig. 2(b) better displays the information we intend to convey. The color scheme helps the reader to observe the energy of the transition, whereby the red represents lower energy and blue shows higher energy. The band nesting can then be observed by the uniformity of the color across the Brillouin zone. In the red-to-black scheme, the band nesting and energy information is less discernible.

These deficiencies can be observed in the red-to-black version shown in Fig. R3. We hope that our explanation is satisfactory.

Fig. R3. Band nesting region of 1L and 2L MoS₂ with a simple red-black color bar

5. Page 9 line 16, “Interlayer coupling can also be minimized by insulting a buffer layer”, insult? insert?

We thank the reviewer for pointing out this typo and we have changed the wording to “inserting” as shown below.

Third paragraph on page 9,

Interlayer coupling can also be minimized by ~~insulting~~ **inserting** a buffer layer...

6. Page 10 line 20, “the thickness of the SiO₂ thickness” ?

We thank the reviewer for pointing out this typo and we have corrected the wording as shown below.

Third paragraph on page 10,

Here, the thickness of the SiO₂ ~~thickness~~ was chosen to be 199 nm...

7. Page 14 line 17 methods part, For the calculations, both *d* and *p* orbitals of Mo should be taken into account. Did the authors compare the cases with and without Mo-*p* orbitals?

We really appreciate the careful review.

For the wannierization, we used *d* orbital for Mo (and also for other transition metal) and *s* and *p* orbital for chalcogenides. As you may know, *p* orbital of transition metal is not important for most transition metals because it is just semicore electron. Among transition metals, we only consider *s* and *p* orbital for Zr because it only has 2 electrons in its *d* orbital. Again, we thank you for your careful review. We corrected our method part as follow:

Computational details section on page 14,

For the wannierization, we used *d* ~~and *p*~~ orbital projections for **most** transition metals, ***s* and *p* orbital projections** ~~for~~ and chalcogenides, and ***s*, *p*, and *d* orbital projection** for Zr, respectively.

8. Graphene is used as a buffer layer, but graphene is a half metal and induces remarkable quenching effect. why graphene is chosen?

We thank the reviewer for the constructive comment. As stated by the reviewer, graphene does induce a strong quenching effect [*AIP Adv.* **6**, 125201 (2016). *ACS Appl. Mater. Interfaces* **8**, 1644–1652 (2016)] and in photoluminescence (PL) measurements, quenching been observed in

the TMD/graphene heterostructures. The considerable PL quenching can be attributed to enhanced recombination of e–h pairs as a result of interaction with the graphene layer. To clarify the cause for the considerably changed optical properties of stacked MoS₂ with graphene, we have performed confocal PL of MoS₂ with and without graphene as buffer layer (Fig. R4). PL measurements of the MoS₂/graphene heterostructure finds a significant quenching of over 50% in intensity when compared to 1L MoS₂.

Fig. R4. Photoluminescence (PL) of spectra of MoS₂ and MoS₂/graphene heterostructures.

However, it is important to point out that our work is concerned with optical absorption and not emission. While it is true that graphene does indeed lead to strong carrier recombination, this process only occurs after absorption.

Nevertheless, it is interesting to consider whether or not the graphene affects the optical absorption in MoS₂. Fig. S11 shows the optical contrast and absorbance of 1L MoS₂, 2L MoS₂ and a MoS₂/Gr heterostructure. By increasing the number of layers from 1L to 2L of MoS₂, the energy of A (B) exciton was slightly redshifted (blueshifted) due to the interlayer coupling. The C exciton peak also shifts and its dependence is even stronger than that of A and B exciton. On the other hand, we note that all of the exciton peaks (A, B, and C) in the MoS₂/Gr heterostructure retain the same positions as those of 1L MoS₂. Due to the negligible interlayer interaction, the graphene is found to interact very weakly with the MoS₂ monolayer. DFT studies have shown that the MoS₂/Gr heterostructure has an interlayer spacing of 3.32 Å and a binding energy of –23 meV/C atom regardless of the adsorption arrangement, indicating a weak interaction between graphene and MoS₂ [*Nanoscale*, **3**, 3883-3887 (2011)].

Another indication of the lack of interaction between graphene and MoS₂ is shown in Fig. S11(b), which shows the absorbance of intrinsic (1L and 2L) MoS₂ and the MoS₂/Gr heterostructure. We confirmed that the absorbance of the MoS₂/Gr heterostructure (red line)

increases by 2.2 % compared to that of 1L MoS₂ (black line) and does not show a C exciton peak shift. This is strong evidence that graphene does not affect the exciton absorbance of MoS₂.

Fig. S11. (a) Optical contrast (b) absorbance of intrinsic (1L and 2L) MoS₂ and MoS₂/Gr heterostructure on glass and Salisbury screen, respectively.

In the future, hexagonal boron nitride (hBN) could be an alternative choice for the buffer layer, but it was not chosen in this study due to the difficulty in preparing large-area single monolayers.

To address this issue clearly, we have added figure R4 and additional discussion in Supplementary Information.

Experimental details section S8 in Supplementary Information

FIG. S11. (a) Microscope image of sample preparation of stacked MoS₂/Gr/MoS₂ heterostructure with 100 μm scale bar. (b) and (c) Local Raman and PL spectra of MoS₂ for each step. (d) Raman spectra of Gr.

Experimental details section S8 in Supplementary Information

Fig. S11 (c) shows PL measurement of MoS₂ in heterostructure, which reveals significant quenching of over 50 % in intensity when compared to 1L MoS₂. This is usually ascribed to charge transfer process. The considerable PL quenching can be attributed to the reduced recombination of e-h pairs through the heterojunction of graphene/MoS₂. However, as shown in Fig S12, it does not affect the light absorption of MoS₂. Fig. S11 (d) show Raman with c-c bonds of graphene: D, G, and 2D peak. The ratio of I_{2D}/I_G is about 1.58, close to 2 of high-quality single-layer graphene.

9. BN and graphene play similar roles in reducing the interlayer coupling and in enhancing optical absorption. So any buffer layer made of any other material also works?

We thank the reviewer for raising this important issue. In our manuscript, we have explored the use of 3 different buffer layers which are graphene, hBN, and ZnSe. As the reviewer points out, all of three buffer layers play a similar role. Furthermore, a buffer layer does not necessarily have to be a 2D material. We can generally say that it is likely that most wide gap insulators (hBN) are applicable as a buffer layer, but semi-metallic (graphene) and semiconductor materials (ZnSe) may also be suitable. In our revised manuscript, we added a few more sentences describing the requirements of buffer layers.

Third paragraph on page 9,

Interlayer coupling can also be minimized by inserting a buffer layer between the TMD layers, as shown in Fig. 3(a). Graphene (Gr) and hexagonal boron nitride (hBN) are used as buffer layer, primarily due to their experimental feasibility. Optical absorption measurements and first principles calculations both confirmed that they indeed served as good buffer layers.

~~We first performed calculations, using either 1L graphene (Gr) or hexagonal boron nitride (hBN) as a buffer layer.~~

10. Are there experimental results from 3L and 4L, to compare with 2H-bilayer and twisted bilayer? And then 3L with interlayer twist angles?

Thank you for raising this important issue. We measured the absorbance of 3L and 4L of MoS₂ on a glass substrate (Fig. R5 (a)) and with Salisbury screen (Fig. R5 (b)). As MoS₂ becomes thicker, the optical contrast (\propto absorbance) continues to increase with gradual redshift, all of which are in good agreement with previous studies [*Nanotechnology*, **27**, 115705, (2016), *Applied Nanoscience*, **11**, 605–610 (2021)]. The maximum optical contrast of twisted 2L MoS₂ and MoS₂/Gr/MoS₂ heterostructure is almost the same as that of 3L MoS₂ with negligible redshift. Note that such a negligible redshift is practically useful in designing Salisbury screen since its optimum thickness of the dielectric layer is the same as that of the monolayer case. We don't have experimental results on twisted 3L and 4L of MoS₂. However, it is naturally deduced that twisted 3L MoS₂ should have higher optical contrast (or real part optical conductivity) than pristine 3L MoS₂, and also should exhibit much weaker redshift.

With the Salisbury screen structure, both 3L and 4L of MoS₂ also exhibit very strong absorbance because their optical conductivity is also very close to the near-perfect light absorbers (NPLAs) condition. ($2.17 \text{ mS} < \sigma' < 3.24 \text{ mS}$) Our DFT results (Fig. R6 (a) and (b)) also exhibit consistent results with experiments. It can be particularly useful to utilize twisted 3 or 4 layers to design a *perfect* light absorber ($A > 99.9\%$ or even higher). The perfect light absorber with Salisbury screen requires a specific value of σ' very close to 2.654 mS, which is not feasible just using pristine few-layer TMDs. The close relation between twist angle and band nesting of TMDs opens a feasible pathway to design “perfect” light absorbers, providing a new perspective on efficient atomically thin optoelectronics.

To clearly address this point in our manuscript, we added both Fig. R5 and Fig. R6 in the Supplementary Information as Figure S17 and S18, and also added additional discussion in Supplementary Information as Section S10.

Fig. R5. (a) Optical contrast of few-layer MoS₂ on a glass substrate and (b) absorbance with Salisbury screen. Insets in (a) and (b) indicate a degree of the redshift of C exciton and maximum absorbance with Salisbury screen, respectively.

Fig. R6. (a) Theoretically calculated real part of optical conductivities and (b) absorbance with Salisbury screen of few-layer MoS₂.

First paragraph on page 12,

These results are a strong confirmation that reducing interlayer coupling is a very efficient and feasible strategy to enhance A in the multilayer TMDs, showing that a simple and cost-effective NPLAs (or “perfect” light absorbers) can be realized by combining 2D materials and a single mirror cavity structure. Additional discussions on the multilayer TMDs are provided in Section S10, Fig. S17, and Fig. S18 in the SI.

Section “ABSORBANCE OF FEW-LAYER MoS₂” in Supplementary information,

Figure S17(a) and (b) show the absorbance of 3L and 4L of MoS₂ on a glass substrate and with Salisbury screen, respectively. As MoS₂ becomes thicker, the optical contrast (\propto absorbance) continues to increase with gradual redshift, all of which are in good agreement with previous studies.^{S10,S11} The maximum optical contrast of twisted 2L MoS₂ and MoS₂/Gr/MoS₂ heterostructure is almost the same as that of 3L MoS₂ with negligible redshift. Note that such a negligible redshift is practically useful in designing Salisbury screen since its optimum thickness of dielectric layer is the same as that of the monolayer case. We don't have experimental results on twisted 3L and 4L of MoS₂. However, it is naturally deduced that twisted 3L MoS₂ should have higher optical contrast (or real part optical conductivity) than pristine 3L MoS₂, and also should exhibit much weaker redshift.

With the Salisbury screen structure, both 3L and 4L of MoS₂ also exhibit very strong absorbance because their optical conductivity is also very close to the near-perfect light absorbers (NPLAs) condition. ($2.17 \text{ mS} < \sigma' < 3.24 \text{ mS}$) Our DFT results (Fig. R6 (a) and (b)) also exhibit consistent results with experiments. It can be particularly useful to utilize twisted 3 or 4 layers to design a *perfect* light absorber ($A > 99.9\%$ or even higher). The perfect light absorber with Salisbury screen requires a specific value of σ' very close to 2.654 mS, which is not feasible just using pristine few-layer TMDs. The close relation between twist angle and band nesting of TMDs open a feasible pathway to design “perfect” light absorbers, providing a new perspective on efficient atomically thin optoelectronics.

[S10] Castellanos-Gomez, A., Queda, J., van der Meulen, H. P., Agra'it, N. & Rubio-Bollinger, G. Spatially resolved optical absorption spectroscopy of single- and few-layer MoS₂ by hyperspectral imaging. *Nanotechnology* **27**, 115705 (2016).

[S11] Neri, I., L'opez-Su'arez, M., Caponi, S. & Mattarelli, M. Fast MoS₂ thickness identification by transmission imaging. *Appl. Nanosci.* **11**, 605–610 (2021).

Fig. S17. (a) Optical contrast of few-layer MoS₂ on a glass substrate and (b) absorbance with Salisbury screen. Insets in (a) and (b) indicate a degree of the redshift of C exciton and maximum absorbance with Salisbury screen, respectively.

Fig. S18. (a) Theoretically calculated real part of optical conductivities and (b) absorbance with Salisbury screen of few-layer MoS₂.

Therefore I cannot support its publication at this stage, considering the limited novelty.

We believe we have addressed all concerns in our revised manuscript, and are convinced that the revised manuscript is significantly improved and now meets the high standard of *Nature Communications*.

Reviewer #2 (Remarks to the Author):

The manuscript by Seungjun Lee et al. presents interesting results of theoretical and experimental work aiming toward achieving near-perfect light absorption in atomically thin transition metal dichalcogenides through band nesting and embedding these materials in an optical resonator. The nesting of valence and conduction bands in TMDCs i.e. existence of parallel in energy bands is visible in results of ab-initio band structure calculation by several groups including the author, resulting in large optical conductivity. The large optical conductivity from Kohn Sham orbitals is also partially supported by BSE-GW methodology. The band nesting is significant for a single layer. For highly absorbing materials one needs to increase the number of layers. Unfortunately as argued by the authors interlayer coupling is reducing band nesting effects. Hence the authors propose to increase the number of layers while reducing their electronic coupling by either twisting two layers or inserting a different material in between the TMD layers. In the end the layers of TMDCs are inserted into Salisbury screen resonator and very high absorptivity is demonstrated.

Overall I find this work to be a very nice application of unique properties of TMDCs – band nesting. The paper is well written and results appear sound. Where the paper could improve is to provide not only observation but also explanation of the physical origin of band nesting. This has been done and as an example I refer authors to two papers: Bieniek et al., “Band nesting, massive Dirac Fermions and Valley Landé and Zeeman effects in transition metal dichalcogenides: a tight binding model”, *Phys.Rev.B* 97, 085153 (2018). and Bieniek, et al., “Band nesting and exciton spectrum in MoS₂”, *Phys.Rev.B* 101, 125423 (2020).

This additional discussion would significantly improve the quality and impact of the manuscript and I would be happy to recommend publication.

We thank the Reviewer for their time and effort in reviewing our manuscript. We are strongly encouraged by the Reviewer's positive and favorable comments. We strongly agree with the suggestion that some additional discussion on the origin of band nesting of TMDs will improve the quality of our manuscript. We added additional discussions in our revised manuscript and cited suggested references, as follow:

Second paragraph on page 5,

Due to the excellent band nesting ($\nabla_k(E_c - E_v) \sim 0$) along Γ -Q and Γ -M high symmetry lines (green arrows), 1L MoS₂ exhibits strong light absorption near 2.81 eV, where the absorption peak at this energy is often referred to as the C exciton.^{22,31} **Studies have shown that the excellent band nesting can be traced to the second nearest neighbor hopping between multiple metal d orbitals, which produces a local minimum in the conduction band at Q valley and thus promotes the nesting of the bands.**^{32,33}

Reference section on page 19,

32 Bieniek, M. et al. Band nesting, massive dirac fermions, and valley landé and zeeman effects in transition metal dichalcogenides: A tight-binding model. *Phys. Rev. B* 97, 085153 (2018).

33 Bieniek, M., Szulakowska, L. & Hawrylak, P. Band nesting and exciton spectrum in monolayer MoS₂. *Phys. Rev. B* **101**, 125423 (2020).

Reviewer #3 (Remarks to the Author):

Authors used band nesting of TMDC materials and Salisbury geometry to achieve NPLA. They did thorough theoretical study on absorbance and optical conductivity of different types of 1L 2D materials. They also provided simulation, detail calculation, material and device fabrication, and characterization. Overall, this piece of work is interesting and could be useful for 2D materials community. However, authors need to address below queries and correct the mistakes.

We thank the reviewer for thier positive comments about the usefulness of our manuscript. Please find below our point-by-point replies.

1. In the introduction, they mentioned that “these complex structures require demanding nano-patterning processes resulting in expensive fabrication costs and are thus limited to applications requiring only small-area structures”. In their twisted 2L MoS₂ fabrication, E-beam patterning is involved. This is a slow process and cannot be expand to large area. How will the authors justify the statement mentioned in introduction?

It is important to point out that our use of E-beam patterning was only performed as a matter of convenience for our research lab. The actual patterns used were relatively large on the order of microns and could have been created using simple optical lithography. In the future, it could even be possible to completely eliminate the need for lithography, since large-area single crystal growth of 2D materials has already been demonstrated [Nat Commun 11, 2153 (2020)]. For this reason, we believe that our technique has significant potential for the development of near-unity light absorbers using a low-cost, large-area-compatible technique.

2. Based on the findings from Fig. 1(c), 2H phase of SnSe₂ has the highest σ' of more than 1mS. Why did the authors choose MoS₂ instead of SnSe₂ to demonstrate?

We thank the reviewer for asking this question. There are a couple of reasons why we did not choose SnSe₂. First of all, to maximize absorbance using Salisbury screen, a *very good* reflector at the target wavelength is necessary. Although SnSe₂ exhibits the highest σ among TMDs, its optimum wavelength is around 4.2 eV. However, conventional metals such as Ag, Au typically absorb the light with energy $> \sim 4$ eV due to the interband transitions. In our experimental setup, we also used Ag, which absorbs light where $\hbar\omega > \sim 3.5$ eV. Therefore, practically, it is difficult to use SnSe₂. In addition, we have found it is more difficult to achieve single monolayers of SnSe₂ compared to MoS₂. Therefore, MoS₂ was chosen demonstrate our theoretical predictions. In the revised manuscript, we added a few sentences describing our choice of experimental materials.

First paragraph on page 5,

Among these TMD materials, the 2H phase of MoS₂ and SnSe₂ exhibit very high σ' of around 1 mS, which is significantly larger than that of graphene. **In particular, monolayer MoS₂ is readily accessible in experiments, in conjunction with the fact that cavity structures in the visible**

can be easily realized. However, unfortunately, they are still lower than the theoretical target for the ultimate Salisbury screen ($2.17 \text{ mS} < \sigma' < 3.24 \text{ mS}$), implying that the realization of NPLA with TMD would require more than one layers.

3. The authors did not discuss on full width at half maximum (FWHM) of respective curves in Fig. 4(c). The twist 2L MoS₂ has the broader spectrum among others. Why?

Fig. R7. Normalized absorbance of 2L MoS₂ and twisted 2L MoS₂ on Salisbury screen

In Fig. 4(c), the FWHM of absorption peaks are determined by not only the FWHM of σ' of each 2D layer but also resonance effects of the cavity. Naively (ignoring oscillation strength), it is intuitive that the FWHM of σ' should have an inverse relation to the degree of band nesting. For example, ideal band nesting leads to a joint density of states (JDOS) in the form of a δ -function, while poor band nesting gives a very broad JDOS spectrum. Therefore, as shown in Fig. 4(c) of the main manuscript, MoS₂/Gr/MoS₂ shows a narrower peak than twisted MoS₂ and 2L MoS₂. The twisted MoS₂ and 2L MoS₂ have the almost same FWHMs, as shown in both theory and experiment. (See Fig. R7) This is due to the cavity resonance effects. Without resonance, as shown in Fig. 2(d) in our main manuscript, twisted MoS₂ have a smaller FWHM than 2L MoS₂. However, the difference in linewidth is minor because it is limited by cavity resonance.

4. When the authors obtained absorbance from different samples with reflection measurement, did they consider scattering effect? Can you include the experiment result to check the scattering responses from your samples?

We thank the reviewer for this excellent question. To minimize scattering effects, we utilized a highly flat SiO₂ layer as the optical spacer by the template stripping method compared to conventional fabrication techniques. Using a sacrificial layer to temporarily bond the thin film to a Si substrate (Fig. S13 in the Supplementary Information), the sacrificial layer helps to maintain the flatness of the substrate during the deposition process, reducing the formation of defects and impurities. Here, we used Au as the sacrificial layer. The reason why template stripping creates a flatter surface is due to fact that the sacrificial layer acts as a buffer, absorbing any surface roughness or defects that may be present on the substrate. As a result, SiO₂ as the optical spacer is able to conform to the roughness of the sacrificial layer, further improving the flatness of the surface.

Fig. R8. AFM images of (a) SiO₂ on Si substrate and (b) SiO₂ using template stripping method. (c) Line profile of cross section of SiO₂.

Fig. R9. Absorbance of MoS₂/Gr/MoS₂ on Salisbury screen in different locations

To verify this, the AFM images were acquired by using the tapping mode AFM system. Fig. R8 (a) shows an AFM image of SiO₂ on a Si substrate. The root-mean-square (RMS) value of the SiO₂ films was 3.07 nm. On the other hand, the SiO₂ by template stripping method has a lower RMS value of 0.98 nm. (Fig. R8 (b)) Both samples have same thickness of 199-nm by PECVD. This result indicates the SiO₂ using template stripping method had a much smoother surface morphology than conventional PECVD films. Fig. R8 (c) shows the height profile of a cross section of SiO₂ by direct deposition (black) on Si substrate and template stripping method (red). Obviously, SiO₂ obtained by the template stripping method exhibits greater uniformity and flatness compared to conventional fabrication process.

Even when measuring various locations on the sample (MoS₂/Gr/MoS₂ on Salisbury screen structure), we observed little variation as show in Fig. R9. This is because we did not need to consider scattering effects when measuring reflectance to obtain the calculated absorption. As we used the template stripping method to obtain a very flat SiO₂ optical spacer in Salisbury screen structure, scattering effects were minimized.

This allowed us to obtain accurate and reliable optical measurements, even when measuring multiple locations on the sample.

To address this point in our revised manuscript, we added Fig. R8 in our supplementary information and also revised main manuscript as follow:

Second paragraph on page 11,

To validate our theoretical prediction by experiments, we fabricated SiO₂(199 nm)/Ti(3 nm)/Ag(90 nm)/Au(60 nm) heterostructure on a glass substrate by the template stripping method (Fig. S14 in SI), which offers a ultra-flat surface morphology. (Fig. S15 in SI) and performed optical measurements. The results are shown in Fig. 4(c). Then, we performed optical measurements, as shown in shown in Fig. 4(c).

Figure S15 in Supplementary Information

Fig. S15. AFM images of (a) SiO₂ on Si substrate and (b) SiO₂ using template stripping method. (c) Line profile of cross section of SiO₂.

5. At page 12, it should be “conclusion” or “summary” instead of “discussion”.

We thank the reviewer for this suggestion, and we revised the section heading accordingly.

6. At page 15, during the fabrication of Salisbury screen structure, a bare silicon wafer without pre-cleaning was used. Do you think it is not necessary? Why?

We do not believe that pre-cleaning was necessary for this demonstration, though in a standard manufacturing process, it is possible that pre-cleaning would be performed. The most important aspect of the wafer preparation is that chemical and particle impurities are removed without altering or damaging the wafer surface. In our process, before the fabrication of the Salisbury screen structure, a bare silicon wafer was first heated on a hot plate at 180 °C for 5 min and then an oxygen plasma surface treatment (Advanced vacuum, Vision 320) was applied for 5 min. O₂ plasma cleaning is often used to clean the surface of the Si wafer before the sacrificial layer is deposited to ensure that it is free of contaminants and has a clean surface for the subsequent deposition steps. This is important for ensuring the quality of the deposited layers and the successful transfer of the thin film to the new substrate. The O₂ plasma cleaning process can

remove organic residues and other contaminants that may be present on the surface of the Si wafer, resulting in a clean and uniform surface for subsequent processing steps. We have mentioned this in the Methods section of the manuscript. We took the motif from our technique known as template stripping [Nat. Nanotechnol. 14, 313 (2019), Nat. Commun. 11, 1 (2020)], which allows the high-throughput fabrication of ultrasmooth surface by replicating them via a reusable silicon template. Our method is to obtain an ultrasmooth surface with only surface treatment by oxygen plasma without additional chemical cleaning of the bare silicon wafer.

This is mentioned in the Method in manuscript as follows:

Fabrication of Salisbury screen structure in Method on page 15

Fabrication of Salisbury screen structure

For Salisbury screen structure with SiO₂ as an optical spacer, a bare silicon wafer was first heated on a hot plate at 180°C for 5 min and then surface treatment by oxygen plasma (Advanced vacuum, Vision 320) for 5 min. The O₂ plasma cleaning process can remove organic residues and other contaminants that may be present on the surface of the Si wafer, resulting in a clean and uniform surface for subsequent processing steps. The silicon wafer was deposited with a 60 nm thick gold film as a sacrificial layer by an electron-beam evaporator (CHA industries, SEC 600).

Supplementary Information on page 19

Fig. S14. Fabrication process; Before the fabrication, a bare silicon wafer was first heated on a hot plate at 180°C for 5 min and then surface treatment by oxygen plasma (Advanced vacuum, Vision 320) for 5 min. 1. Gold (Au) deposition with a 60 nm thick as a sacrificial layer by an electron-beam evaporator (CHA industries, SEC 600).

7. For Salisbury screen structure, what is the tolerance of each layer thickness such as 199 nm of SiO₂, 90nm of Ag and 60nm of Au?

We thank the reviewer for the question. The SiO₂ dielectric spacer was deposited with a thickness of 199 nm by plasma-enhanced chemical vapor deposition (Plasma-Therm 790, PECVD). The SiO₂ deposition rate was 5.7 Å/sec at a stage temperature of 100 °C. The thickness variation of the film was less than ±3 % across a 4" wafer. For the Salisbury screen structure, spectroscopic ellipsometry was performed to confirm the thickness of the 199-nm-thick SiO₂ after Au deposition on the wafer. (Fig. R7(a)) Fig. R9 (b) shows the results of thickness and uniformity measurements for the PECVD-grown SiO₂ on top of Au on the Si wafer. The average thickness from 345 measurement locations was 198.78 nm with a standard deviation of 0.49 nm. Although there were deviations in the SiO₂ thickness deposited by PECVD on the wafer, we selectively used only 199-nm-thick region to fabricate our structures.

The Ag and Au films were deposited at 90 nm and 60 nm thick, respectively, by an electron-beam evaporator with a deposition rate of 1 Å/sec. Light in the visible (350 – 850 nm) region is reflected from Ag mirror at close to % and the change in reflection properties for the Ag mirror has a negligible dependence on thickness variations of the metal films.

Fig. R10 (a) optical micrograph showing after deposition 60 nm of Au (e-beam evaporation) and 199 nm of SiO₂ (PECVD) on the 4" wafer. (b) Ellipsometry measurements showing the thickness uniformity of the PECVD-grown SiO₂.

8. During characterization of samples, did the author take into consideration of polarization effect especially on twisted sample?

The operation principle of Salisbury screen simply relies on the destructive interference between reflective light from the surface and metal reflector. Therefore, our device structure, Salisbury screen, is not sensitive to the polarization of incident light, and thus we did not consider polarization effects in our experiments.

Note that, recently, it was reported that twisted 2D materials exhibit a strong circular dichroism, which means that its absorbance depends on the handedness of circular-polarized incident light.

[*Nat. Nanotech.* **11**, 520–524 (2016)]. This effect arises due to the in-plane magnetic moment associated with the interlayer optical transition. However, this effect also should be negligible in our device structure, since the incident light should pass through the twisted bilayer twice when it comes in and out. Therefore, although the circular dichroism of twisted bilayer MoS₂ is an interesting topic, this effect is not important in our Salisbury screen structure. Most importantly, we used normal incident unpolarized light in experiment, so any polarization effects due to circular dichroism will be averaged to null anyway.

9. The indication of PMMA and Au is not correct at step 4 of Fig S4 (a).

We thank the reviewer for pointing out the mislabeling of layers in our figure. To make it clear, we have modified Fig S4 as follows.

FIG. S5. (a) Schematic illustration of the process of Au-assisted mechanical exfoliation. (b) Optical microscopy image of the large-scale MoS₂ on Au film after the peeling-off process. (c) Optical (left) and microscope (right) image of MoS₂ on Si/SiO₂ substrate.

10. In page 8 of supporting information, there is typo error. “tow” should be “two”.

We thank for reviewer for this correction and we have revised the manuscript as follows:

Supplementary Information on page 8

The finger print region in the Raman spectrum of 1L MoS₂ exhibits ~~tow~~two main modes E_{12g}¹ at 385 cm⁻¹ and A_{1g} at 404 cm⁻¹ with mode difference of 19 cm⁻¹.

11. In page 14 of supporting information, the abbreviation of graphene should be “Gr” instead of “gr”.

We thank for reviewer for this correction and we have revised the manuscript as follows:

Supplementary Information on page 14

PMMA(MicroChem, 950 PMMA C4) was spin-coated on CVD-grown Gr on Cu foil at 3,000 r.p.m for 1 min and baked at 180 °C for 15 min.

12. In S9, the authors used 192nm of SiO₂, 3 nm of Titanium and 100 nm of Silver to form Salisbury screen structure. Do all layers need to have the exact thickness as mentioned? Why is there layers difference in this Salisbury screen structure from MoS₂ sample?

For the MoS₂ experiments, since the C exciton is 2.83 eV, SiO₂ was deposited at 199 nm to achieve the critical coupling condition. On the other hand, for the WSe₂/ZnSe/WSe₂ structures, the B' exciton of WSe₂ is located at 2.92 eV. Based upon numerical simulations, the SiO₂ thickness was chosen to be 192 nm, satisfying the critical coupling condition for this heterostructure. This demonstrates the advantage of our technique, since by simply adjusting the thickness of SiO₂ for the Salisbury screen structures, it is possible to improve the optical absorption in the band nesting for a wide class of 2D materials.

Here, titanium (Ti) was used as an addition layer to increase adhesion at the interface between silver (Ag) and SiO₂, however, the thickness of the Ag and the precise nature of the underlaying layers of the metal stack are not critical for the Salisbury screen.

We have revised the manuscript as follows:

Supplementary Information on page 17

To make the optical cavity, the sample was taken out of the MBE, moved to a plasma enhanced chemical vapor deposition (PECVD) chamber, and capped with 192 nm of SiO₂ deposited at 250°C. Here, thickness of SiO₂ was chosen to be 192 nm, satisfying the critical coupling condition for the photon energy of 2.92 eV of B' exciton of WSe₂.

Reviewers' Comments:

Reviewer #1:

Remarks to the Author:

All the concerns raised have been addressed. Pay attention to the typo throughout the manuscript and the incomplete sentence on Page 7, "For the similar reason, we also do not include SOC in what follows because it does not have critical effects on our main result; the relation between band nesting and interlayer interaction. (Fig. S4 in the SI.)"

Reviewer #2:

Remarks to the Author:

The Authors successfully addressed my comments and improved their manuscript. Publication is recommended.

Reviewer #3:

Remarks to the Author:

Authors replied all queries from reviewers by attempting in point-to-point and corrected the mistakes. I am happy with their responses.

However, there are still two minor mistakes.

1. In the response to Q7 of Reviewer #3, there is missing number "is reflected from Ag mirror at close to % and..." at the second last line of your reply.
2. The indication of PMMA and Au is still not correct at step 4 of Fig S5(a) in the revised supporting information. If it is difficult to indicate correctly, I would suggest to use legends with notations instead of direct writing on every step.

Dear Dr. Silvia Milana,

We are very pleased to hear that our manuscript has been accepted. All the comments raised by the editorial office were incorporated into our revised manuscript. We thank you for taking care of the review process of our manuscript, and all the referees for their constructive comments and suggestions.

REVIEWER COMMENTS

Reviewer #1 (Remarks to the Author):

All the concerns raised have been addressed. Pay attention to the typo throughout the manuscript and the incomplete sentence on Page 7, "For the similar reason, we also do not include SOC in what follows because it does not have critical effects on our main result; the relation between band nesting and interlayer interaction. (Fig. S4 in the SI.)"

We indeed thank you for your effort in reviewing our manuscript and recommendation for publication. Your review is indeed helpful to improve our manuscript. We carefully proofread our manuscript again and corrected typos in the final version of manuscript.

Reviewer #2 (Remarks to the Author):

The Authors successfully addressed my comments and improved their manuscript. Publication is recommended.

We indeed thank you for your effort in reviewing our manuscript and recommendation for publication!

Reviewer #3 (Remarks to the Author):

Authors replied all queries from reviewers by attempting in point-to-point and corrected the mistakes. I am happy with their responses. However, there are still two minor mistakes.

We thank you for your effort in reviewing our manuscript and recommendation for publication. Your review is indeed helpful to improve our manuscript.

1. In the response to Q7 of Reviewer #3, there is missing number "is reflected from Ag mirror at close to % and..." at the second last line of your reply.

We are sorry for this typo. It should be 100%. The correct sentence is, "Light in the visible (350 – 850 nm) region is reflected from Ag mirror at close to 100 % and the change in reflection

properties for the Ag mirror has a negligible dependence on thickness variations of the metal films.”

2. The indication of PMMA and Au is still not correct at step 4 of Fig S5(a) in the revised supporting information. If it is difficult to indicate correctly, I would suggest to use legends with notations instead of direct writing on every step.

We updated Fig. S5(a) in the final version of Supporting information. Thank you very much your careful proofreading.

FIG. S5. (a) Schematic illustration of the process of Au-assisted mechanical exfoliation. (b) Optical microscopy image of the large-scale MoS₂ on Au film after the peeling-off process. (c) Optical (left) and microscope (right) image of MoS₂ on Si/SiO₂ substrate.